# The Toxicological Assessment of *Anoectochilus burmannicus* Ethanolic-Extract-Synthesized Selenium Nanoparticles Using Cell Culture, Bacteria, and *Drosophila melanogaster* as Suitable Models

**DOI:** 10.3390/nano13202804

**Published:** 2023-10-22

**Authors:** Pensiri Buacheen, Jirarat Karinchai, Woorawee Inthachat, Chutikarn Butkinaree, Chonchawan Jankam, Ariyaphong Wongnoppavich, Arisa Imsumran, Teera Chewonarin, Nuttaporn Pimpha, Piya Temviriyanukul, Pornsiri Pitchakarn

**Affiliations:** 1PhD Program in Biochemistry, Faculty of Medicine, Chiang Mai University, Chiang Mai 50200, Thailand; pensiri_bua@cmu.ac.th; 2Department of Biochemistry, Faculty of Medicine, Chiang Mai University, Chiang Mai 50200, Thailand; jirarat.ka@cmu.ac.th (J.K.); ariyaphong.w@cmu.ac.th (A.W.); arisa.bonness@cmu.ac.th (A.I.); teera.c@cmu.ac.th (T.C.); 3Food and Nutrition Academic and Research Cluster, Institute of Nutrition, Mahidol University, Nakhon Pathom 73170, Thailand; woorawee.int@mahidol.ac.th (W.I.); piya.tem@mahidol.ac.th (P.T.); 4National Omics Center, National Science and Technology Development Agency, Pathum Thani 12120, Thailand; chutikarn.but@nstda.or.th (C.B.); chonchawan.jan@ncr.nstda.or.th (C.J.); 5National Nanotechnology Center, National Science and Technology Development Agency, Thailand Science Park, Pathum Thani 12120, Thailand; nuttaporn@nanotec.or.th

**Keywords:** selenium, orchid, nanotechnology, *Drosophila*, wing spot test, bacterial reverse mutation test, *Anoectochilus burmannicus*, Ames test, nanoparticle, novel food, toxicity testing

## Abstract

Selenium nanoparticles (SeNPs) are worthy of attention and development for nutritional supplementation due to their health benefits in both animals and humans with low toxicity, improved bioavailability, and controlled release, being greater than the Se inorganic and organic forms. Our previous study reported that *Anoectochilus burmannicus* extract (ABE)-synthesized SeNPs (ABE-SeNPs) exerted antioxidant and anti-inflammatory activities. Furthermore, ABE could stabilize and preserve the biological activities of SeNPs. To promote the ABE-SeNPs as supplementary and functional foods, it was necessary to carry out a safety assessment. Cytotoxicity testing showed that SeNPs and ABE-SeNPs were harmless with no killing effect on Caco2 (intestinal epithelial cells), MRC-5 (lung fibroblasts), HEK293 (kidney cells), LX-2 (hepatic stellate cells), and 3T3-L1 (adipocytes), and were not toxic to isolated human PBMCs and RBCs. Genotoxicity assessments found that SeNPs and ABE-SeNPs did not induce mutations in *Salmonella typhimurium* TA98 and TA100 (Ames test) as well as in *Drosophila melanogaster* (somatic mutation and recombination test). Noticeably, ABE-SeNPs inhibited mutation in TA98 and TA100 induced by AF-2, and in *Drosophila* induced by urethane, ethyl methanesulfonate, and mitomycin c, suggesting their anti-mutagenicity ability. This study provides data that support the safety and anti-genotoxicity properties of ABE-SeNPs for the further development of SeNPs-based food supplements.

## 1. Introduction

Selenium nanoparticles (SeNPs), nano-sized elemental selenium particles, have been developed and widely studied for biomedical applications because of their unique properties including low toxicity with better bioavailability and biological activities compared with the traditional selenium supplementation (inorganic and organic forms) [1,2]. Recently, SeNPs have not only been used in selenium-deficient individuals but have also been of great interest as a therapeutic agent for the prevention or treatment of metabolic disorders and diseases. SeNPs exhibit potential therapeutic applications for humans such as anti-oxidant, anti-cancer, anti-inflammation, and anti-diabetes as well as antimicrobial activity that can be applied in various fields, including infection control, biomedical instrument surface treatment, pharmaceutical industry, cosmetics, and food manufacture. [3,4,5,6,7,8]. Equally important, SeNP supplements in livestock feed can improve production, growth, feed efficiency, antioxidant status, and immune status compared to inorganic selenium [9,10]. Furthermore, SeNPs contained in feed can effectively provide a high content of selenium in animal products including egg, milk, and meat [11] that can support the sufficient intake of selenium in humans through the food chain.

Normally, SeNPs are synthesized through a chemical reduction reaction using reducing agents such as ascorbic acid and glutathione [12]. In addition, plant extracts that contain polyphenol compounds and have antioxidant abilities are candidates for use as reducing agents and stabilizers for SeNP synthesis, known as the green synthesis method [13]. For example, *Emblica officinalis* fruit extract, rich in phenolics, flavonoids, and tannins, is responsible for both the reduction and stabilization of nano-selenium synthesis [14]. Our previous study reported the antioxidant and anti-inflammatory activities of SeNPs using *Anoectochilus burmannicus* ethanolic extract (ABE). ABE acts as a stabilizer that prevents physical property changes and improves the biological activities of the suspended SeNPs. In addition, it functions as a cryoprotectant and/or lyoprotectant during the freeze-drying process of the SeNPs, causing the particles to be completely resuspended while maintaining their physical and biological features [15]. Even though multiple studies are showing various biological activities that support green synthesized nano-selenium to be used as a functional food, its toxicity and safety evaluation remains limited.

Since biologically synthesized metallic nanoparticles, especially using plant extract, have been a major focus for novel food or ingredient development, their toxicological study is needed to assess and certify them for safe use. The European Food Safety Authority (EFSA) has provided new guidance on the risk assessment of nanomaterials for application in the food and feed chain, and human and animal health [16]. In vitro testing followed by in vivo verification has been recommended to be applied for toxicity and genotoxicity risk assessment. An in vitro cytotoxicity/cell viability test is a general method that is used to determine and screen the toxicity of compounds/agents/materials to cells. According to the nanomaterials entering via an oral route through the GI tract (mouth, stomach, intestine) and translocating to other organs such as the lung, liver, and kidney via blood circulation [17], then the EFSA guidance suggests that the toxicity of the nanoparticles should be carried out in at least two or three different cell types which are represented in the above-mentioned organs and also in immune cells such as macrophages.

Genotoxicity is a property of physical, chemical, and biological agents that damage the genetic information within a cell resulting in DNA damages and genomic instabilities which may lead to cancer [18]. Genotoxicity and carcinogenicity have been occasionally found in various types of biogenic nanoparticles as well as in natural product supplements, suggesting that genotoxic assessment is necessary for novel food development to guarantee food safety use in humans and animals [19,20].

The bacterial reverse mutation assay or Ames test is the primary in vitro test for genotoxicity screening which determines the potential mutagenic activity of xenobiotics (e.g., drugs, chemical agents, pesticides) using *Salmonella typhimurium*, the bacterial histidine-dependent strain [21]. The mutant *S. typhimurium* cannot grow because of its histidine synthesis ability, but if a possible mutagenic substance causes a reverse mutation, then the bacteria retrieve its ability to synthesize histidine and can grow on a histidine-free medium [22]. The increasing revertant colony number indicates the possibility of the suspicious samples to be mutagenic agents. In addition, some agents are pro-mutagens that require metabolic activation to become mutagens through oxidation with cytochrome P450 (CYP450) [23]. To mimic mutagenesis via the CYP450 metabolism, the mouse liver homogenate is applied in the Ames test [21]. However, the Ames test can detect limited types of DNA mutations, such as point mutations and frameshift mutations; therefore, a parallel in vivo test must be conducted.

The somatic mutation and recombination test (SMART) or wing spot test using *Drosophila melanogaster* (fruit fly) is a well-known model for food safety assessment [24]. It can be an alternative in vivo model for genotoxicity evaluation; 60% of the *Drosophila* genome is homologous to humans, and about 75% of the genes causing human diseases have homologs in flies [25,26]. Using the *Drosophila* model is useful and convenient because of their short life cycle, large numbers of embryos, and easy genetic modification [27]. In addition, adult and larva flies have CYP450-dependent activation systems able to metabolize most pro-mutagens which is applicable to determine the genotoxicity of suspicious compounds [28]. Importantly, SMART can be used to detect wide ranges of DNA damages, including point mutations and DNA breaks, as well as mitotic recombination [29,30].

As the safety assessment of nano-selenium is necessary for supporting SeNPs-based food supplement development, this study, therefore, investigated the cytotoxicity of ABE-SeNPs and its core materials (Se, SeNPs, and ABE) in five cell lines including human lung fibroblasts (MRC-5), human embryonic kidney cells (HEK293), human hepatic stellate cells (LX-2), mouse adipocyte fibroblasts (3T3-L1), and human colon carcinoma cells (Caco2), as well as in isolated human primary cells, human peripheral blood mononuclear cells (hPBMCs), and red blood cell (hRBCs). Furthermore, the genotoxicity study of the particles including mutagenic and anti-mutagenic activities was performed using a bacterial reverse mutation assay and *Drosophila* wing spot test.

## 2. Materials and Methods

### 2.1. Plant Sample and Extraction

*Anoectochilus burmannicus* (*A. burmannicus*) sample tissue was cultivated and obtained from the Queen Sirikit Botanic Garden, Chiang Mai, Thailand. The sample was identified and deposited in the Queen Sirikit Botanic Garden, Chiang Mai, Thailand. The herbarium voucher specimen is S. Watthana 4494 (QBG). The ethanolic extract (ABE) was prepared as previously described [31]. Briefly, a dried whole plant of *A. burmannicus* was powdered and soaked in 80% ethanol at the ratio of 10:100 *w*/*v* with occasional shaking and stirring. The ethanolic fraction was filtered and evaporated using a rotary evaporator at a temperature of 50 °C. The yield of ABE was about 25% (*w*/*w*). The ABE powder was kept at −20 °C until used.

### 2.2. Total Phenolic Content, HPLC Profile, and Kinsenoside Content of A. burmannicus Ethanolic Extract (ABE)

Total phenolic content was measured using the colorimetric method using Folin–Ciocalteu reagent in accordance with the previous study [31]. Briefly, 300 μL of the extracts was added to 400 μL of 10% equivalent Folin–Ciocalteu reagent and incubated for 3 min in the dark at room temperature. Then, 300 μL of 7.5% equivalent/L Na_2_CO_3_ was added and the mixture was allowed to stand for 20 min in dark, then the absorbance was measured at 763 nm using a spectrophotometer. Total phenolic content was expressed as mg gallic acid or ferulic acid/g extract equivalent to the standard curves of gallic acid and ferulic acid in ABE, respectively. 

Phytochemical profile of ABE was further determined with HPLC, using a C18 column (250 × 4.6 mm, 5 μm) (Agilent Technologies, Santa Clara, CA, USA). The chromatographic separation was carried out using a gradient system of mobile phase A (1% acetic acid in water) and mobile phase B (100% acetonitrile) with a total run time of 50 min for detection with a flow rate of 0.7 mL/min. The gradient system used was 90% A in 0 min–60% in 28 min, followed by 40% in the next 39 min and 10% in the next 50 min. The extract of 10 mg/mL dissolved in 1 mL of MeOH was injected into the column and detection at 280 in phytochemical contents. The peak area and retention time of the extract sample were evaluated as the comparison with the curve of standards including gallic acid, chlorogenic acid, caffeic acid, ellagic acid, rosmarinic acid, quercetin, kaempferol, catechin, mangiferin, vanillic acid, rutin, ferulic acid, luteolin and apigenin, and quercetin and kaempferol. The results are shown in the Appendix A). 

Kinsenoside content was analyzed using the LC/MS-MS system. The calibration standards were prepared with final concentrations of 50, 150, 300, 625, 1250, 2500, 5000, and 10,000 ng/mL for kinsenoside. ABE was solubilized in 50% acetonitrile/water at a concentration of 1 mg/mL. Acetonitrile containing 0.1% acetic acid (250 µL) was added to the sample, mixed, and then centrifuged at 15,000× *g* for 10 min at 4 °C. Thereafter, 200 µL of supernatant was transferred to a clean microcentrifuge tube then dried using centrifugal concentrator (TOMY, Tokyo, Japan). The dried extracts were reconstituted in 100 µL of 50% acetonitrile/water and submitted into the LC-MS/MS analysis. The UltiMate 3000 UHPLC system (Thermo Fisher Scientific, Waltham, MA, USA) equipped with an Atlantis HILIC Silica (2.1 mm × 100 mm I.D. 3.0 µm, Waters, Wilmslow, UK) with column temperature at 30 °C. The mobile phase consisted of solvent A (5 mM ammonium acetate in water, pH 3.0) and solvent B (acetonitrile) and was flowed at 300 µL/min. The total run time was 7 min for each sample. Samples were injected for 15 µL and eluted at the following condition: 0–2 min, 90%B; 2–5 min, 50%B; 5–7 min, 90%B. Eluted samples were then subjected to mass spectrometer, SCIEX Triple Quad 5500+ (AB SCIEX, Concord, ON, Canada), for analysis. Multiple reaction monitoring (MRM) scan type was acquired with ESI source performed in a positive ion mode. A quantification of kinsenoside with the ion transitions monitored (*m*/*z*): 265.0 → 102.9 for KD. The ion source temperature and ion spray voltage were set as 600 °C and 5500 V. The gas 1 and gas 2 were set as 16 and 30 psi. The declustering potential values were both set as 60 V for kinsenoside and IS. The collision energy was 46 V for KD.

### 2.3. Preparation of A. burmannicus Ethanolic-Extract-Synthesized Selenium Nanoparticles (ABE-SeNPs)

ABE-SeNPs were synthesized as previously described [15]. Briefly, 0.1 M sodium selenite (Na_2_SeO_3_) solution (0.3 mL) was mixed with 10 mg/mL of ABE (28.5 mL) and stirred at 600 rpm for 10 minutes. The solution of 0.1 M ascorbic acid (1.2 mL), initiator for the reduction step, was added dropwise into the mixture and stirred for six hours. The nanoparticles were purified through dialysis against Milli-Q water at 4 °C for 24 h to remove unreacted materials including the extract, ascorbic acid, and Na_2_SeO_3_. Moreover, SeNPs were also synthesized using the same method, but ABE was replaced with an equal volume of Milli-Q water. The brick-red-color solution represents the success of SeNP synthesis.

### 2.4. Characterization of A. burmannicus Ethanolic-Extract-Synthesized Selenium Nanoparticles (ABE-SeNPs) 

Size and zeta potential were determined using dynamic light scattering analysis (DLS, Ultro Pro, Malvern, Worcestershire, UK) with a Zetasizer Nano instrument. Each sample was dried and then ground into a homogeneous powder to record the infrared spectra on a Nicolet iS50 FT-IR spectrometer (Thermo Fisher Scientific, Waltham, MA, USA)with a built-in diamond-attenuated total reflection (ATR). The spectra were acquired at 400–4000 cm^−1^ wavenumbers with a 4 cm^−1^ resolution with sample scanning 32 times. The concentration of the nanoparticle occurred through inductively coupled plasma analysis (ICP-OES hydride generation, Perkin Elmer, 7300 DV, MA, USA) [15].

### 2.5. Determination of Kinsenoside Content in A. burmannicus Ethanolic-Extract-Synthesized Selenium Nanoparticles (ABE-SeNPs)

ABE-SeNPs were extracted with methanol (ratio 1:1) to obtain ABE, the mixture then was centrifuged at 12,000 rpm for three minutes. Thereafter, 200 µL of supernatant was transferred to a clean vial and used for LC-MS/MS analysis, as described in Section 2.2. The calibration standards were prepared with final concentrations of 50, 150, 300, 625, 1250, 2500, 5000, and 10,000 ng/mL for kinsenoside.

### 2.6. Cell Line and Cell Culture

Four normal cell lines MRC-5 (human lung fibroblast cells, passage number ranged from 10 to 15), HEK293 (human embryonic kidney cells, passage number ranged from 10 to 20), LX-2 (human hepatic stellate cells, passage number ranged from 10 to 20), 3T3-L1 (murine adipocytes, passage number ranged from 5 to 10), and Caco2 (human colon carcinoma cells, passage number ranged from 30 to 40) which are represent lung, kidney, liver, adipose tissue, and intestinal epithelium, respectively, were obtained from American Type Culture Collection (ATCC, Manassas, VA, USA). Despite being malignant, the Caco2 cell line is extensively utilized as an in vitro model of the intestinal epithelial barrier [32,33]. The cells were cultured as adherent cells in a cell culture flask in Dulbecco’s Modified Eagle Medium (DMEM) with L-glutamine supplement containing 10% fetal bovine serum (FBS) (MRC-5, HEK293, LX-2, and Caco2) or 10% fetal calf serum (FCS) (3T3-L1) and 1% penicillin under 5% CO_2_ at 37 °C. When the cells grew to 80% cell confluency, the cells were collected and subjected to experimentation.

Human peripheral blood mononuclear cells (hPBMCs) and red blood cells (hRBCs) were derived from human blood samples as previously described by Karinchai et al. [31]. Human blood samples of healthy subjects were obtained from Maharaj Hospital, Chiang Mai, Thailand, which were anonymized by the laboratory and kept in a heparinized tube. PBMCs were isolated using Ficoll-Hypaque separation, according to the manufacturer’s instructions [34]. The remaining RBCs were used in a hemolysis assay. This study was granted a certificate of exemption by the Research Ethics Committee of the Faculty of Medicine, Chiang Mai University (No. EXEMPTION 8554/2023). Consent to participate was not applicable because of anonymous data collection. All methods were carried out following relevant guidelines and regulations.

### 2.7. Cytotoxic Assay

The five cell lines, 3T3-L1 (3 × 10^3^ cells/well), HEK-293 (5 × 10^3^ cells/well), Caco2 (5 × 10^3^ cells/well), MRC-5 (4.5 × 10^3^ cells/well), LX-2 (6 × 10^3^ cells/well), and isolated hPBMCs (8 × 10^4^ cells/well) were seeded in a 96-well plate and treated with various doses of SeNPs, ABE-SeNPs, or Na_2_SeO_3_ (4–16 µM of selenium concentration) for 24 and 48 h. To determine cytotoxicity in the cell lines, an MTT assay was performed, as previously described [35]. The treated cells were incubated with 0.5 mg/mL MTT solution with serum-free medium for two hours (formazan crystal formation). The solution was gently removed, then the crystal was dissolved with dimethyl sulfoxide (DMSO). The cell viability is directly related to the absorbance of formazan. The absorbance of the colored solution was measured at 540 nm with a microplate reader.

The cytotoxicity of the nano- and elemental Se on hPBMCs was determined using an SRB assay, as described previously [15]. In brief, the treated cells were fixed and stained with SRB dye for 30 min, and the stained cells were washed with 1% (*v*/*v*) acetic acid to remove excess dye. The protein-bound dye was dissolved in 10 mM Tris-base solution for the determination of absorbance at 510 nm using a microplate reader.

The percentage of cell viability was calculated using the following equation.
Cell viability (%) = (Absorbance (sample))/(Absorbance (control)) × 100

### 2.8. Hemolysis Assay

A hemolysis assay was used to determine the hemolytic effect of SeNPs, ABE-SeNPs, and Na_2_SeO_3_. Human RBCs were collected after the separation of hPBMCs from whole blood. The 5% RBCs suspension then was mixed with several concentrations of the various forms of selenium and incubated at 37 °C for three hours. The hemolysis induction was measured as described previously [36]. Hemoglobin released by damaged cells can be used as a marker of red blood cell lysis following test agent exposure and can notice as a red-color solution that can be measured for absorbance using spectrophotometry. Triton X-100 was used as a positive control. The hemolytic effect of the samples was indicated following the guideline that when <10% hemolysis is non-hemolysis and >25% hemolysis is high hemolysis induction [37].

### 2.9. Genotoxicity Assessment Using Bacterial Reverse Mutation Assay (Ames Test)

Ames test was performed following the guidelines of OECD 471 [38]. Mutagenic and antimutagenic effects of SeNPs, ABE-SeNPs, Na_2_SeO_3_, and ABE were performed as previously described [39].

To investigate the mutagenicity of the samples, *S. typhimurium* strains TA98 and TA100 were mixed with various concentrations of SeNPs, ABE-SeNPs, Na_2_SeO_3_ (3.12–12.5 µM), or ABE (1–4 mg/mL) in the presence and absence of S9 mixture, a metabolic activation. 2-aminoanthracene (2-AA) and 2-(2-furyl)-3-(5-nitro-2-furyl)-acrylamide (AF-2) standard mutagens were used as positive control. 2-AA was used in the presence of S9 mixture, while AF-2 was used without the S9 mixture. Deionized water (DI) and DMSO were used as negative controls for the samples and mutagens, respectively. The mixture solution was poured onto the minimal histidine agar and then incubated at 37 °C for 48 h. The revertant colonies formed on the agar plate were scored. The revertant colony number of samples that were two times higher than spontaneous revertant colonies (negative control) in a dose-dependent manner was supposed to be a mutagen.

To determine the antimutagenicity of the samples, 6-phenylimidazo [4,5-b]pyridine (PhIP) and 2-Amino-3-methylimidazo [4,5-f]quinoline (IQ) were used as standard mutagens (positive control) in the test of TA98 and TA100, respectively, in the presence of S9 mixture. The antimutagenicity was indicated based on percentage inhibition which was calculated as in the following equation:Antimutagenicity (%) = [1 − (b/a)] × 100

a = revertant colony number induced via standard mutagen (positive control)b = revertant colony number induced by standard mutagen in the presence of each sample treatment

Three plates were investigated per dose in two independent experiments for both mutagenicity and antimutagenicity tests.

### 2.10. Survival Assay in Drosophila melanogaster

To evaluate the non-toxic concentration of the nano-selenium for further study in *Drosophila*, a survival assay was performed following Inthachat et al. [40]. In brief, two strains of *Drosophila* which are *mwh* strains, containing wing cell marker multiple wing hair (*mwh*) and the ORR strain (*flr3/In (3LR) TM3*, *ri p^p^ sep l(3)89Aa bx^34e^ eBd^s^*), containing chromosomes 1 and 2 from DDT-resistant Oregon R(R) line were mated to produce trans-heterozygous larvae expressing cytochrome P450 enzymes involved in xenobiotic biotransformation (F1 progenies). Then, 100 F1 larvae were fed on the regular medium containing SeNPs, ABE-SeNPs, or water (DI). After 7 days, the number of hatched flies was recorded and the survival rate was calculated and compared to DI. The non-toxic concentrations were used for further experiments.

### 2.11. Genotoxicity Assessment of ABE-SeNPs and SeNPs in Drosophila

The mutagenicity and anti-mutagenicity of ABE-SeNPs and SeNPs were performed using the Somatic mutation and recombination test (SMART) or wing spot test following Inthachat et al. [40]. To investigate the mutagenic potential of the nanoparticles, 100 three-day-old larvae (third larvae stage) derived from Section 2.10 were collected and transferred to a *Drosophila* medium with a non-toxic concentration of ABE-SeNPs or SeNPs. Urethane (20 mM) was used as a mutagen (positive control) and DI was used as a negative control. The larvae were fed on each medium until pupation at 25 °C. Later, the wings of surviving flies were removed for mutant spot analysis. Approximately 40 wings in each group were scored for the presence of mutant spots, including small single spots, large single spots, and twin spots. The results were diagnosed using the estimation of spot frequencies and confidence limits for comparison with the control group, and the statistical significance was calculated as previously described by Frei and Würgler [41].

To determine the anti-mutagenicity of the sample, 100 three-day-old larvae derived from Section 2.10 were fed with *Drosophila* medium containing mutagens, which are 20 mM urethane, 1 mM ethyl methanesulfonate (EMS) or 0.1 mM mitomycin C (MMC) with or without SeNPs or ABE-SeNPs (1–20 µM) treatment until the pupation. The wing spots were scored as mentioned above. The percentage of anti-mutagenicity of the nanoparticles (% inhibition) was calculated as in the following equation:Antimutagenicity (%) = [(a − b)/a] × 100

a = number of total spots per wing of positive mutagen control groupb = number of total spots per wing of each experimental group

The *Drosophila* study was approved by the Institute of Nutrition-Mahidol University Institutional Animal Care and Use Committee (INMU-IACUC) (COA.No. INMU-IACUC, 2023/02).

### 2.12. Statistical Analysis

All values were given as mean ± standard derivation (X ± SD) from triplicate samples of three independent experiments. Overall, the differences among the treatment groups were determined using a one-way analysis of variance (ANOVA), followed by Tukey’s multiple comparison test using Prism 9.0 software. *p* values < 0.05 were regarded as a measure of statistical significance. For the wing spot test, statistical analysis was calculated following Frei and Würgler (1988) with significance levels of α = β = 0.05 [41].

## 3. Results

### 3.1. Characterization of A. burmannicus Ethanolic-Extract-Synthesized Selenium Nanoparticles (ABE-SeNPs) and Kinsenoside Content

As a quality control of the plant sample and the extraction, the total phenolic content of ABE was 11.18 ± 0.74 mg GAE/g extract and 23.28 ± 1.09 mg FAE/g extract which was equivalent to the ABE used in our previous studies [31,42] (Appendix A). The HPLC analysis of ABE found the phenolic compounds including catechin, mangiferin, rutin, ferulic acid, luteolin, and apigenin (Appendix A). In addition, kinsenoside content in ABE analyzed using LC-MS was 371 ± 36.75 µg/mg extract.

As shown in Table 1, the physical properties including size, polydispersity index (PDI), and zeta potential as well as FTIR chromatogram of SeNPs and ABE-SeNPs (Appendix A) were comparable with our previous report [13], suggesting that the nanoparticle synthesis is reproducible.

Kinsenoside is one of the major active compounds in *Anoectochilus* sp. [43] and exerts antioxidant, anti-hyperglycemia, anti-inflammation abilities [44,45]. Moreover, our previous study reported that the kinsenoside found in *A. burmannicus* exhibits anti-inflammation and anti-obesity activities [31,42]. It therefore can be used as a standard marker of the orchid extract. To ensure that ABE is incorporated with the SeNPs, the kinsenoside content of ABE-SeNPs was subsequently measured. The LC/MS-MS showed that the sharp peak of kinsenoside was detected in ABE-SeNPs, whereas the peak was not observed in SeNPs (Appendix A), suggesting that ABE was surely incorporated into SeNPs. Kinsenoside content in ABE-SeNPs was 793.5 ± 33.23 µg/mL. Moreover, the amount of ABE that participated in SeNPs could be calculated wtih the standard curve of kinsenoside. The result revealed that the kinsenoside content of ABE-SeNPs was comparable to that of 61 mg of ABE (Initial weight of ABE = 285 mg), indicating that ABE at approximately 21.4% was incorporated into SeNPs.

### 3.2. Cytotoxicity and Hemolytic Induction Property of the SeNPs and ABE-SeNPs

Nanoparticles can interact with proteins in the blood or cells in organs when delivered to the human body, which may be harmless or damage the cells and tissues [46]. Therefore, the cytotoxicity of the nano-selenium was screened in different cell lines including Caco2, MRC-5, HEK293, LX-2, and 3T3-L1, which represented intestinal epithelium, kidney, liver, lung, and adipose tissue, respectively, as well as in hPBMCs and hRBCs which are cellular components of blood. Figure 1 shows that different chemical forms of selenium had varying effects on the viability of each cell type, inhibitory concentration at 50% cell viability or the half-maximal inhibitory concentration (IC50) of SeNPs was over 16 µM in all cell types except Caco2. The IC50 of ABE-SeNPs was 7 µM in HEK-293 and 3T3-L1 and over 16 µM in MRC-5 and LX-2, whereas IC50 of Na_2_SeO_3_ was 9, 10.5, and 11.2 µM in HEK-293, MRC-5, and LX-2, respectively. The compound at up to 16 µM was not toxic to pre 3T3-L1. Among five cell lines, Caco2 was mostly affected by sodium selenite, SeNPs, and ABE-SeNPs with IC50 at 4, 4.6, and 3.6 µM, respectively. Caco2 is a colon carcinoma cell line that is widely used as a model of the intestinal epithelial barrier. It might be possible that Se was specifically more toxic to cancer cells than normal cells. Remarkably, ABE-SeNPs seem to be more toxic than sodium selenite and SeNPs in 3T3-L1, HEK-293, and Caco2, which are highly proliferative cell types. It might be possible that ABE-SeNPs inhibited cell proliferation rather than induced cell death. The cell number of ABE-SeNP-treated cells was increased compared to the starting point of the treatment. In addition, the growth of the treated cells was slowly increased when compared to the control. Moreover, under microscopic observation, the morphology of the cells remained unchanged, suggesting that ABE-SeNPs inhibited the proliferation of 3T3-L1, HEK-293, and Caco2 (Appendix A) leading to a decrease in cell numbers. Meanwhile, sodium selenite showed a killing effect on HEK-293, Caco2, MRC-5, and LX-2 represented by the reduction in the cell number after the treatment (48 h) compared to the starting point (0 h) (Appendix A).

Three forms of selenium did not affect the cell viability of hPBMCs (at up to 16 µM of Se) (Figure 1F) and did not induce hemolysis of hRBCs (at up to 50 µM of Se) (Table 2). Taken together, the results revealed that the cytotoxicity effect of selenium was dependent on its chemical form and variable based on exposed cell type. However, the killing effect was not observed in both forms of selenium nanoparticle, suggesting the low toxicity of the nano-selenium compared with the elemental form.

### 3.3. Mutagenicity of SeNPs and ABE-SeNPs in S. typhimurium and D. melanogaster

The mutagenicity and anti-mutagenicity of Na_2_SeO_3_, ABE, SeNPs, and ABE-SeNPs were performed with the Ames test using *S. typhimurium* carrying the gene mutation of histidine operon, which cannot grow in minimal histidine medium. The mutagens can induce DNA mutations in the bacteria, thereby restoring the ability to synthesize histidine. The mutagen-induced *S. typhimurium* can therefore grow in histidine-free media, as evidenced by the formation of a revertant colony. Since there are two types of mutagens, direct and indirect (pro) mutagens, the latter of which require metabolic activation (liver S9 homogenate) to biotransform into mutagens. We then performed the Ames test with or without metabolic activation to cover both types of mutagens. The toxicity of each sample was screened to avoid false negative results from the killing effect. The used concentrations of selenium (3.12–12.5 µM) and ABE (1.0–4.0 mg/mL) were non-toxic to the bacteria. The mutagenic potential of the tested samples is demonstrated in Table 3. Two positive controls, including AF-2 and 2-AA, were used as direct and indirect mutagens, respectively. The results revealed that the number of revertant colonies did not differ between treatments and the negative control in both TA98 and TA 100 strains in the presence or absence of S9 homogenate, implying that sodium selenite, ABE, SeNPs, and ABE-SeNPs were neither pro-mutagen nor mutagen. To confirm and support the safe use of the nano-selenium, especially ABE-SeNPs, as functional food, the genotoxicity of SeNPs and ABE-SeNPs were further investigated in the next experiment using *Drosophila melanogaster*.

*Drosophila* (fruit fly) is one of the most well understood of all the model organisms with various gene homologs in mammals. Thus, the genotoxicity of the nano-selenium was assessed via SMART assay in *Drosophila* as an in vivo assay. SeNPs and ABE-SeNPs (at up to 20 µM of Se) were not toxic to *Drosophila*, as shown in Figure 2. Urethane, a mutagen, was used as a positive control to induce somatic cell mutation in larvae. The mutation can be observed through the appearance spots (single and twin spots) of adult wings (Figure 3). As shown in Table 4, There was no difference in the frequency of mutant spots between larvae fed with SeNPs or ABE-SeNPs (4–20 µM of Se) and the negative control, where urethane-exposed larvae exhibited a significant induction of mutant spots, indicating that SeNPs and ABE-SeNPs did not induce mutation in the organism. Taken together, these results suggest that SeNPs and ABE-SeNPs as well as the core materials including sodium selenite and ABE have no mutagenesis ability.

### 3.4. Anti-Mutagenicity of SeNPs and ABE-SeNPs in S. typhimurium and D. melanogaster

To investigate the anti-mutagenic activity of the nano-selenium, the food mutagens, PhIP, and IQ were used as standard mutagens in TA98 and TA100, respectively, in the presence of S9, whereas AF-2 was used as a standard mutagen in both strains in the absence of S9 mixture. Table 5, ABE at 4 mg/plate slightly inhibited PhIP- and IQ-induced mutagenesis in TA98 and TA100, respectively, at approximately 10%. Sodium selenite significantly decreased the number of revertant colonies by 10% (12.5 µM/plate) in PhIP-treated TA98 and 18–21% (6.25–12.5 µM/plate) in IQ-treated TA100 compared to the control. The anti-mutagenic activity of ABE and sodium selenite was not observed in the AF-2-induced mutation model either in TA98 or TA100. In the TA98 strain, SeNPs significantly inhibited PhIP- and AF-2-induced mutation by 10% (6.25–12.5 µM/plate) and by 17–19% (6.25–12.5 µM/plate), respectively. In the TA100 strain, SeNPs exhibited an anti-mutagenic effect against AF-2, but not IQ to induce mutation by 28–38% (6.25–12.5 µM/plate). Interestingly, ABE-SeNPs significantly suppressed mutagen-induced reverse mutation in TA98 and TA100 both with or without metabolic activation (S9 mix) conditions. ABE-SeNPs decreased mutagen-induced revertant colonies in TA98 and TA100 in the presence of S9 by 12–13% (6.25–12.5 µM/plate) and 6–14% (6.25–12.5 µM/plate), respectively. Moreover, ABE-SeNPs significantly inhibited AF-2-induced revertant colonies both in TA98 and TA100 by 16–31% and 20–48% (3.12–12.5 µM/plate), respectively. The results suggest that the formulation of Se as the nano-selenium could improve its anti-mutagenic activity. Moreover, the synthesis of the nano-selenium with ABE (ABE-SeNPs) could enhance the anti-mutagenic ability compared to bare SeNPs.

Interestingly, SeNPs and ABE-SeNPs (4–20 µM of Se) exerted anti-mutagenicity ability in *Drosophila* as shown by the frequency of mutant spots induced by urethane, EMS, and MMC, which were reduced in the treatment groups (Table 6). Urethane-induced mutations were slightly inhibited by SeNPs (5–9%) and were effectively inhibited by ABE-SeNPs (16–19%). EMS activity was weakly suppressed by SeNPs (5–9%), while the inhibitory effect of ABE-SeNPs (7–15%) on EMS-induced mutant spots was higher than that of SeNPs. The MMC-induced mutation model for both forms of nano-selenium were similarly effective. SeNPs and ABE-SeNPs (4–20 µM of Se) moderately inhibited MMC-induced mutation at approximately 24–30% and 11–28%, respectively. Taken together, the results suggested that SeNPs and ABE-SeNPs did not induce mutation and exerted anti-genotoxicity ability. Moreover, the fabrication of SeNP with ABE can increase the anti-genotoxicity property of the particles in urethane- and EMS-induced mutagenicity models, but not in MMC-treated organisms. The efficiency of SeNPs and ABE-SeNPs may be following the class of mutagens and their mechanism of action.

## 4. Discussion

Selenium (Se) is an essential trace element that plays an important role in human health and animal growth. Se supplements are used for the prevention or treatment of selenium deficiency. In the majority of supplements, selenium is present as selenomethionine (organic form), whereas sodium selenite and sodium selenate (inorganic form) are predominantly used in livestock feed [47]. In recent decades, selenium has significantly gained attention in the prevention or treatment of metabolic disorders and diseases including atherosclerosis [48], insulin resistance [49], and cancer [50]. However, long-term or excessive intake as well as some chemical forms of selenium may cause toxicity to humans [51,52]. Likewise, the overdose of selenium in animal feed results in acute or chronic selenosis, known as an alkali disease leading to weakness and mortality [53]. Then, selenium nanoparticles (SeNPs) were developed to be therapeutic drugs and Se nutritional supplements because of their unique properties, low toxicity, high bioactivity, and gradual release which overcome a limitation of the traditional Se. In addition, biogenic SeNP synthesis and its biological activity have been widely studied in the medical field, especially the synthesis using plant extract containing high phenolic and flavonoid because it can be used as a reducing and/or stabilizing agent and also enhance bioactivity. Although abundant data supports the therapeutic application of biogenic SeNPs, their safety assessment is still limited.

*A. burmannicus* contains high phenolic compounds and polysaccharide components as well as kinsenoside which are promising bioactive components of *Anoectochilus* sp. [31,43,54,55]. According to our previous study, *A. burmannicus* extract (ABE)-synthesized selenium nanoparticles (ABE-SeNPs) exert biological activities including anti-oxidant and anti-inflammation. Moreover, ABE improves the stability that preserves physical and biological properties and prolongs shelf life by functioning as a cryo or lyoprotectant of SeNP powder [15]. These data encourage us to further develop ABE-SeNPs as a SeNP-based functional food. As there are several factors including size distribution, morphology, surface charge, surface chemistry, and capping agents that could impact the toxicity of the nanoparticles [56]; safety assessment is required to understand the possible adverse effects and their fate inside the human body. Following the EFSA guidelines [16], we performed a cytotoxicity test, a screening tool for toxicity assessment, of various selenium forms, SeNPs, ABE-SeNPs, and sodium selenite in different cell lines involved in the absorption system and destination organs. Our results found that a group of highly proliferative cell types (3T3-1, HEK-293, and Caco2) was sensitive to ABE-SeNPs more than those of slow-cycling cells (MRC-5 and LX-2) (Figure 1). In addition, ABE-SeNPs were not toxic to primary hPBMC, the cells were non-proliferative, indicating that ABE-SeNPs inhibited cell proliferation rather than induced cell death, as confirmed by the growth curve analysis. Sodium selenite was toxic to HEK-293, Caco2, MRC-5, and LX-2 with a killing effect at high concentrations (16 µM) but was not toxic to 3T3-L1 and hPBMCs compared to SeNPs and ABE-SeNPs. These data suggest the cytotoxicity effect of selenium was dependent on its chemical form and variable based on exposed cell types. Consistently, Galić et al. revealed that not only target cells but also the surface functionalization of SeNPs influence their toxic profile. They found that when SeNPs were modified with stabilizers that present a cationic group on the particle surface, the particles would be more harmful to human cell lines which represent biological barriers of entry including human buccal epithelial squamous carcinoma (TR146), human keratinocytes (HaCaT), and human colon carcinoma (Caco2). The particles showed the highest toxicity in TR146 followed by HaCaT and Caco2 [33]. Similarly, Boroumand et al. reported that chitosan-coated SeNPs, which exhibit positive surface charge, showed higher toxicity than negative surface charge polyvinyl-alcohol-coated SeNPs in murine fibroblast L929 cell [57]. Abbas et al. also showed that the variable toxicity of biogenic SeNPs was dependent on the cell types. SeNPs biosynthesized using Spirulina platensis were more harmful in normal human kidney (Vero) cells compared to normal human liver (THLE-2) cells [58].

Notably, three forms of Se which are sodium selenite, SeNPs, and ABE-SeNPs were highly toxic to Caco2 at the lowest concentration of Se (IC50 ≈ 4 µM) compared to the other cells. Although Caco2 is a human colorectal adenocarcinoma, it has been widely used as an intestinal epithelial barrier in an in vitro model because of its properties with spontaneous differentiation into the enterocyte-like cells of the small intestine [59]. The data could indicate that Se was less toxic and much safer in normal cells compared to cancer cells, which are similar to previous studies of Salem et al. and Vahidi et al. [60,61].

Additionally, the EFSA guidelines advise evaluating any potential negative effects of the nanoparticle on immune cells. Following that, we conducted the cytotoxicity testing of sodium selenite, SeNPs, and ABE-SeNPs on human peripheral blood mononuclear cells (hPBMCs), which are the main cells involved in the body’s immune system’s defense against pathogens and contain a variety of immune cells such as lymphocytes, monocytes, or macrophages [62]. It was found that Se was not harmful to hPBMCs in any of its three forms. After being absorbed through the intestine, the nanoparticles travel through the blood circulation to the organs. We then investigated the toxic effect of the Se on red blood cells (RBCs), a major bloodstream cell type via cellular damage measurement or hemolysis assay. Hemolysis was not induced by any of the Se dosages and forms used in the present study. These findings may point to Se’s safety for human use. However, additional animal testing to verify their safety is required.

In addition to cytotoxicity evaluation, genotoxicity assessment is crucial to determining the potential harm of the nanoparticles. The Ames test (OECD TG 471), a bacterial reverse mutation assay, is one of the most widely used in vitro studies in toxicology to assess the possible mutagenic effect of test substances by using bacteria. Mutagens can directly bind to DNA and form covalent bonds or transfer an alkyl group to a nucleotide base resulting in DNA adducts that can lead to mutation and eventually cancer. In addition, some agents are pro-mutagens that must undergo metabolic activation by CYP450 before becoming active mutagens [63,64]. Following the absence of a metabolic activation system in bacteria, an S9 mixture isolated from the liver of Swiss albino mice is used as exogenous metabolic activation to improve the potentiality of bacterial test systems [21]. Various synthetical or environmental mutagens were used as positive controls in the genotoxicity experiment. AF-2 was an antimicrobial food additive in Japan but it was banned since its carcinogenicity was found in mice [65]. Then, it has been used as a positive control to approach the genotoxicity and carcinogenicity assessment of suspicious agents [66,67,68]. Aromatic amines including 2-aminoanthracene (2-AA) are most widely used as pro-mutagen in the Ames assay [69,70]. 2-AA is initially N-hydroxylated by CYP1A1/2 and further catalyzed into the metabolite that can bind to DNA [71,72]. Aromatic amines can be found in daily life such as tobacco smoking, a variety of heat-processed foods with high protein content, and also precursors of pharmaceutical products [69,73]. Likewise, heterocyclic amines are known to be foodborne mutagens usually found in cooked meat (e.g., pork, beef, chicken, and fish) such as 6-phenylimidazo [4,5-b]pyridine (PhIP) and 2-Amino-3-methylimidazo [4,5-f]quinoline (IQ) which are ranked as a possible mutagen (Group 2A) and probable mutagen (Group 2B) in human, respectively, and there has been reported mutagenesis in Ames test and carcinogenesis in rodents [74,75]. In this study, we screened the mutagenic activity of nano-selenium using *S. typhimurium* strain TA98 and TA100 represented frameshift mutation and base pair substitution mutation, respectively. When compared to 2-AA and AF-2, standard mutagens, sodium selenite, ABE, and both types of selenium nanoparticles, SeNPs, and ABE-SeNPs, did not exhibit the mutagenic effect, indicating that the nano-selenium and its core materials were neither pro-mutagen nor mutagen. Furthermore, selenium in nanoform, in particular ABE-SeNPs, demonstrated greater anti-mutagenicity against non-metabolically activated mutagen (AF-2) than pro-mutagens (PhiP and IQ). Usually, genotoxicity assessment in the in vivo model is required to verify the result from the in vitro study, following the limitations of metabolic activation and lack of digestive system. Transgenic rodent somatic and germ cell gene mutation assay (OECD TG 488), one of the mammalian in vivo models, is extensively performed to identify the possibility of genotoxic substances [76]. Currently, genotoxicity assessment in other organisms is promising to reduce the use of mice or rats. Thus, we evaluated the genotoxicity of the nanoparticle using the wing somatic mutation and recombination test (SMART) in *D. melanogaster*.

The SMART or wing spot test in *Drosophila* based on the loss of heterozygosity (LOH) is caused by genetic damage during wing disc cell division in the larvae stage resulting in abnormal hair phenotype on adult wings corresponding to two recessive wing cell marker genes: multiple wing hairs (*mwh*) and flare (*flr*) [77]. The *mwh* phenotype is characterized by close-growing multiple hairs, typically two to four, of various lengths, and the *flr* phenotype is indicated by a single shortened and amorphous hair [24,78]. Mutations caused by mutagens may result in many different forms, such as point mutations, deletions, chromosome loss, and mitotic recombination, which are visible as mutant spots or hair spots on the wings. In the present study, we employed three different types of mutagens to study the antigenotoxicity of SeNPs and ABE-SeNPs. Urethane, also known as ethyl carbamate, is a Group 2A compound present in fermented foods and alcoholic drinks. It is converted by CYP450 into the mutagen vinyl carbamate epoxide (VCE), which interacts with DNA and causes deletion and point mutations [79]. Mitomycin C (MMC), a chemotherapy drug, requires metabolic activation to be active and induce chromosome recombination caused by interstrand crosslinks [80], while ethyl methanesulfonate (EMS) is an alkylating agent that results in methylated nucleotides [81]. The mutagenicity of SeNPs and ABE-SeNPs was not detected in high-bioactivation (HB) flies and this finding was consistent with the result of the Ames test. Interestingly, both forms of the nanoparticles exerted anti-mutagenicity. Although the nanoparticles effectively suppressed mutagenesis against a direct-acting mutagen in the bacteria, they significantly inhibited both pro-mutagen and mutagen-induced genotoxicity in *Drosophila*. While MMC activity was comparably suppressed by SeNP and ABE-SeNPs, ABE-SeNPs exhibited a higher inhibitory effect than SeNPs in the urethane- and EMS-treated flies. According to a study by Sega et al. that demonstrated Artepillin C’s protective impact against MMC but not EMS to induced genotoxicity, the anti-mutagenesis of the nanoparticles may be influenced by the kind of mutation or by the activation of the DNA repair system [82].

Together, the evidence points to the possibility that ABE-SeNPs, when used at an effective dose, may be non-cytotoxic and safe for both humans and animals. However, the high-concentration usage of the nanoparticles may raise concerns due to their impact on cell proliferation. Moreover, the nanoparticles possessed a putative anti-mutagenic property while being free of genotoxicity. To ensure the safety of nanoparticle use in animals and humans, additional research is required, including in vivo or animal models and human clinical trials, as well as the pharmacokinetics of the nanoparticle, including absorption, bioavailability, distribution, excretion, and clearance, etc.

## Figures and Tables

**Figure 1 nanomaterials-13-02804-f001:**
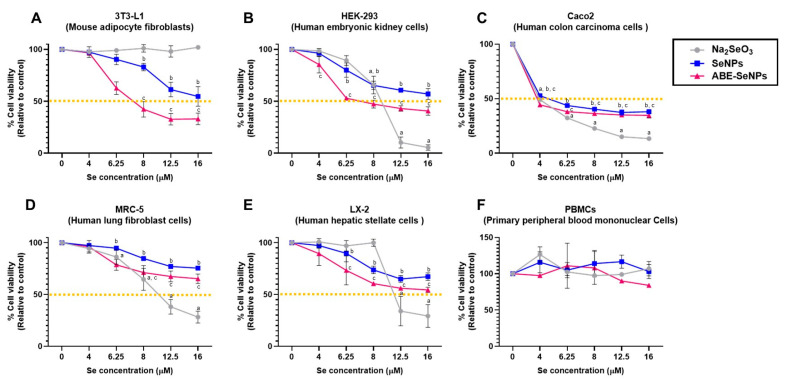
Cytotoxicity of sodium selenite, SeNPs, and Se ABE-SeNPs in 3T3-L1 cell determined via MTT and SRB assay (**A**), HEK-293 cell (**B**), Caco2 (**C**), MRC-5 (**D**), LX-2 (**E**), and PBMCs (**F**). The data are indicated as mean ± SD of three independent experiments. The differences among the treatment groups were determined using a one-way analysis of variance (ANOVA), followed by Tukey’s multiple comparison. *p* values < 0.05 were regarded as a measure of statistical significance. ^a^ Na_2_SeO_3_vs. untreated control, ^b^ SeNPs vs. untreated control, and ^c^ ABE-SeNPs vs. untreated control.

**Figure 2 nanomaterials-13-02804-f002:**
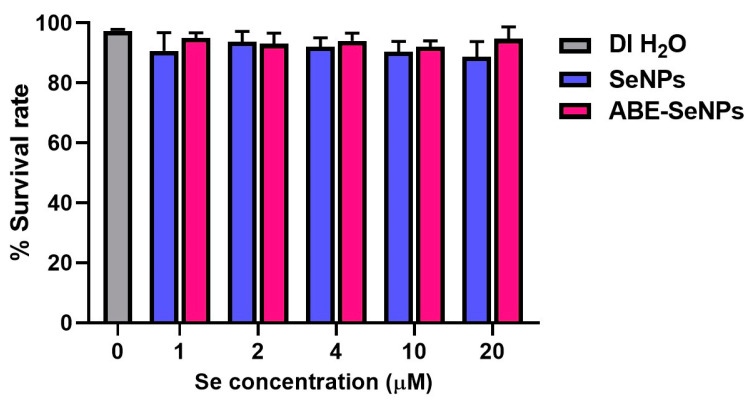
Percentage of *Drosophila* survival rate cultured on various concentrations of SeNPs or ABE-SeNPs (1–20 µM) compared to DI H_2_O (negative control). The data are indicated as mean ± SD of three independent experiments. The differences among the treatment groups were determined using a one-way analysis of variance (ANOVA), followed by Tukey’s multiple comparison. *p* values < 0.05 were regarded as a measure of statistical significance.

**Figure 3 nanomaterials-13-02804-f003:**
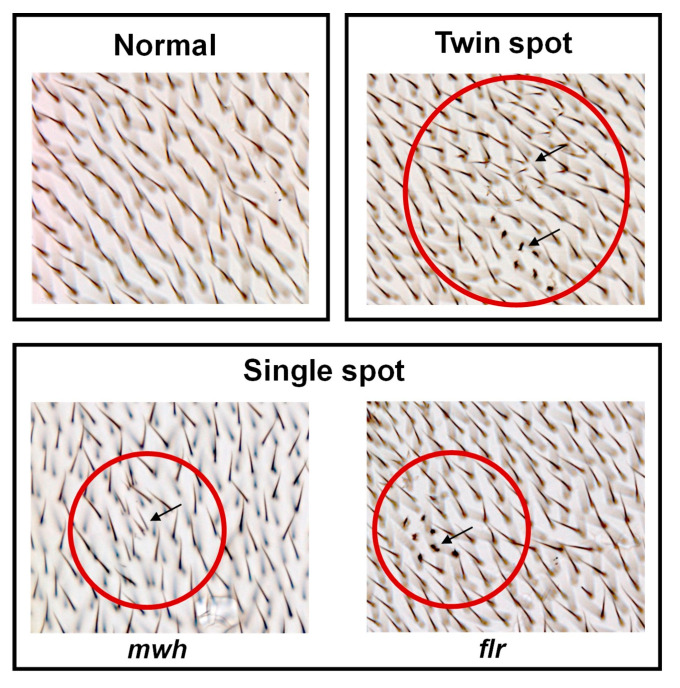
Characteristics observed on the *Drosophila* wings. Characteristics of normal wing spots (each hair in the adult wing is the result of the cuticular secretion of each wing disc cell) (**top left** panel). Characteristics of mutant wing spots (black arrow) (single spots (**bottom** panel) and twin spots (**top right** panel)), which can be induced by genotoxins. mwh—multiple wing hairs, and flr—flare. The mutant spot area is indicated by the arrow in the red circle.

**Table 1 nanomaterials-13-02804-t001:** Size, PDI, and zeta potential of SeNPs and ABE-SeNPs.

	Size (nm)	PDI	Zeta (mV)
SeNPs	97.1 ± 4.24	0.069 ± 0.03	−31.26 ± 0.37
ABE-SeNPs	145.21 ± 2.72	0.353 ± 0.07	−24.01 ± 2.51

**Table 2 nanomaterials-13-02804-t002:** Hemolysis of Na_2_SeO_3_, ABE, SeNPs, and ABE-SeNPs in hRBCs.

Se Concentration (µM)	% Hemolysis		
	Na_2_SeO_3_	SeNPs	ABE-SeNPs
Triton X-100 (positive control)	98 ± 0.052	100 ± 0.23	100 ± 0.23
0	ND	ND	ND
4	0.01 ± 0.059	0.71 ± 0.33	0.91 ± 0.32
6.25	0.14 ± 0.036	0.64 ± 0.37	0.95 ± 0.30
8	0.06 ± 0.024	0.65 ± 0.35	1.02 ± 0.22
12.5	ND	0.66 ± 0.27	1.05 ± 0.57
16	ND	0.65 ± 0.30	0.90 ± 0.39
25	ND	0.46 ± 0.26	0.99 ± 0.39
50	0.08 ± 0.074	ND	0.63 ± 0.18

The results are expressed as mean ± SD, n = 5, 0–10% hemolysis is classed as non-hemolysis, 10–25% hemolysis is slight hemolysis induction, and >25% hemolysis is high hemolysis induction. ND = not detectable.

**Table 3 nanomaterials-13-02804-t003:** The mutagenicity of Na_2_SeO_3_, ABE, SeNPs, and ABE-SeNPs in the presence and absence of metabolic activation (S9 mix) using *Salmonella typhimurium* strains TA 98 and TA 100.

Treatments	Number of Revertant Colonies ^a^			
	TA 98		TA 100	
	+S9	−S9	+S9	−S9
DMSO	29 ± 3	23 ± 4	122 ± 7	112 ± 9
H_2_O	27 ± 3	24 ± 5	118 ± 4	113 ± 8
2-AA (0.5 µg/plate)	318 ± 24 *	N/A	590 ± 38 *	N/A
AF-2 (0.1 µg/plate)	N/A	299 ± 20 *	N/A	500 ± 73 *
ABE				
1 mg/plate	21 ± 2	21 ± 2	119 ± 7	116 ± 5
2 mg/plate	23 ± 3	23 ± 3	120 ± 6	116 ± 3
4 mg/plate	23 ± 6	22 ± 2	127 ± 6	123 ± 5
Na_2_SeO_3_				
3.12 µM/plate	25 ± 5	22 ± 2	121 ± 8	117 ± 7
6.25 µM/plate	26 ± 4	21 ± 1	122 ± 11	110 ± 4
12.5 µM/plate	27 ± 5	23 ± 3	126 ± 7	112 ± 7
SeNPs				
3.12 µM/plate	27 ± 2	24 ± 3	132 ± 9	124 ± 8
6.25 µM/plate	23 ± 2	21 ± 1	133 ± 6	118 ± 8
12.5 µM/plate	27 ± 3	24 ± 2	129 ± 6	122 ± 7
ABE-SeNPs				
3.12 µM/plate	25 ± 2	20 ± 3	131 ± 7	122 ± 10
6.25 µM/plate	26 ± 5	22 ± 1	133 ± 6	123 ± 8
12.5 µM/plate	26 ± 3	22 ± 2	137 ± 5	121 ± 4

^a^ The data are indicated as mean ± SD of spontaneous revertant colonies of triplicate plates per treatment. N/A = not applicable. The differences among the treatment groups were determined using a one-way analysis of variance (ANOVA), followed by Tukey’s multiple comparison. * *p* < 0.05 compared to DMSO.

**Table 4 nanomaterials-13-02804-t004:** Mutagenicity of SeNPs and ABE-SeNPs reported as wing spot tnduction on *Drosophila melanogaster* derived from trans-heterozygous *mwh+/+flr3* larvae of the improved high bioactivation cross.

Treatments	Frequency of Mutant Spots Per Individual (Number of Spots) ^a^			
	Small Single (m = 2)	Large Single (m = 5)	Twin (m = 5)	Total Spots (m = 2)
DI (negative control)	0.53 (21)	0.03 (1)	0.00 (0)	0.55 (22)
Urethane				
20 mM	24.75 (990) +	10.50 (420) +	1.0 (40) +	36.25 (1450) +
SeNPs				
1 µM	0.43 (17) −	0.00 (0) −	0.00 (0) −	0.43 (17) −
2 µM	0.55 (22) −	0.00 (0) −	0.00 (0) −	0.55 (22) −
4 µM	0.68 (27) i	0.00 (0) −	0.03 (1) −	0.70 (28) i
10 µM	0.50 (20) −	0.03 (1) i	0.00 (0) −	0.53 (21) −
20 µM	0.43 (17) −	0.03 (1) i	0.00 (0) −	0.45 (18) −
ABE-SeNPs				
1 µM	0.28 (11) −	0.00 (0) −	0.03 (1) i	0.30 (12) −
2 µM	0.75 (30) i	0.00 (0) −	0.00 (0) −	0.75 (30) i
4 µM	0.60 (24) −	0.00 (0) −	0.00 (0) −	0.60 (24) i
10 µM	0.38 (15) −	0.00 (0) −	0.00 (0) −	0.38 (15) −
20 µM	0.30 (12) −	0.00 (0) −	0.03 (1) i	0.33 (13) −

^a^ Statistical diagnosis using the estimation of spot frequencies and confidence limits according to Frei and Würgler [41] for comparison with DI (negative control); + = positive; − = negative; i = inconclusive. One-sided statistical test “m” is an increased mutation frequency compared with the spontaneous frequency (m times). Number of wings = 40.

**Table 5 nanomaterials-13-02804-t005:** Anti-mutagenicity of Na_2_SeO_3_, ABE, SeNPs, and ABE-SeNPs in the presence and absence of metabolic activation (S9 Homogenate) using *Salmonella typhimurium* strains TA 98 and TA 100.

Treatments	Number of Revertant Colonies ^a^			
	TA 98		TA 100	
	+S9	−S9	+S9	−S9
	PhIP (0.1 µg/Plate)	AF-2 (0.1 µg/Plate)	IQ (0.05 µg/Plate)	AF-2 (0.01 µg/Plate)
Standard mutagens	369 ± 20 (0) ^b^	347 ± 20 (0)	490 ± 20 (0)	460 ± 36 (0)
ABE				
1 mg/plate	339 ± 15 (8)	345 ± 7 (1)	497 ± 39 (−1)	589 ± 35 (−28)
2 mg/plate	337 ± 20 (8) *	345 ± 11 (1)	452 ± 27 (8)	547 ± 21 (−19)
4 mg/plate	331 ± 17 (10) **	354 ± 15 (−2)	437 ± 56 (11)	590 ± 19 (−28)
Na_2_SeO_3_				
3.12 µM/plate	341 ± 14 (7)	315 ± 6 (9)	435 ± 24 (11)	541 ± 82 (−17)
6.25 µM/plate	345 ± 9 (6)	335 ± 24 (4)	404 ± 20 (18) ***	570 ± 40 (−24)
12.5 µM/plate	332 ± 23 (10) *	329 ± 14 (5)	385 ± 28 (21) ***	597 ± 58 (−30)
SeNPs				
3.12 µM/plate	334 ± 11 (9)	327 ± 29 (6)	462 ± 31 (2)	429 ± 62 (3)
6.25 µM/plate	333 ± 14 (10) *	288 ± 30 (17) **	456 ± 34 (3)	320 ± 46 (28) ***
12.5 µM/plate	330 ± 14 (10) **	282 ± 27 (19) ***	446 ± 39 (5)	276 ± 25 (38) ***
ABE-SeNPs				
3.12 µM/plate	336 ± 12 (9)	291 ± 17 (16) **	473 ± 23 (−1)	354 ± 46 (20) *
6.25 µM/plate	325 ± 20 (12)	262 ± 19 (25) ***	440 ± 24 (6)	268 ± 57 (39) ***
12.5 µM/plate	322 ± 22 (13) *	239 ± 23 (31) ***	403 ± 38 (14) *	232 ± 56 (48) ***

^a^ The data are indicated as mean ± SD of spontaneous revertant colonies of triplicate plates per treatment. ^b^ Values in brackets indicate % mutagenic inhibition. The differences among the treatment groups were determined using a one-way analysis of variance (ANOVA), followed by Tukey’s multiple comparison. * *p* < 0.05, ** *p* < 0.01, *** *p* < 0.001 compared to the standard mutagen of each sample treatment.

**Table 6 nanomaterials-13-02804-t006:** Anti-mutagenicity of SeNPs and ABE-SeNPs on urethane, EMS, or MMC-induced mutation in *Drosophila melanogaster* derived from trans-heterozygous *mwh+/+flr3* larvae of improved high bioactivation cross.

Treatments	Frequency of Mutant Spots Per Individual (Number of Spots) ^a^				Inhibition (%)
	Small Single (1–2 Cells)	Large Single (>2 Cells)	Twin Spots	Total Spots	
DI (negative control)	0.425 (17)	0.050 (2)	0.000 (0)	0.475 (19)	−
Urethane					
20 mM	18.425 (737)+	5.400 (216)+	0.800 (32)+	24.625 (985)+	−
SeNPs					
4 µM	17.250 (690)+	5.425 (217)+	0.625 (25)+	23.300 (932)+	5.38
10 µM	16.825 (673)+	4.900 (196)+	0.675 (27)+	22.400 (896)+	9.04
20 µM	16.700 (668)+	5.025 (201)+	0.750 (30)+	22.475 (899)+	8.73
ABE-SeNPs					
4 µM	17.525 (701)+	5.075 (203)+	1.200 (48)+	23.800 (952)+	3.35
10 µM	16.325 (653)+	3.025 (121)+	0.525 (21)+	19.875 (795)+	19.29
20 µM	16.800 (672)+	3.300 (132)+	0.425 (17)+	20.525 (821)+	16.65
EMS					
1 mM	7.400 (296)+	2.850 (114)+	1.050 (42)+	11.300 (452)+	−
SeNPs					
4 µM	7.100 (284)+	2.975 (119)+	0.650 (26)+	10.725 (429)+	5.09
10 µM	7.275 (291)+	2.450 (98)+	0.800 (32)+	10.525 (421)+	6.86
20 µM	7.050 (282)+	2.675 (107)+	0.550 (22)+	10.275 (411)+	9.07
ABE-SeNPs					
4 µM	6.630 (265)+	2.950 (118)+	0.925 (37)+	10.500 (420)+	7.08
10 µM	6.980 (279)+	2.630 (105)+	0.600 (24)+	10.200 (408)+	9.73
20 µM	5.600 (224)+	3.300 (132)+	0.650 (26)+	9.550 (382)+	15.49
MMC					
0.1 mM	9.450 (378)+	0.875 (35)+	0.425 (17)+	10.750 (430)+	−
SeNPs					
4 µM	6.975 (279)+	0.775 (31)+	0.350 (14)+	8.100 (324)+	24.65
10 µM	7.775 (311)+	0.750 (30)+	0.375 (15)+	8.900 (356)+	17.21
20 µM	6.500 (260)+	0.600 (24)+	0.375 (15)+	7.475 (299)+	30.47
ABE-SeNPs					
4 µM	8.350 (334)+	0.750 (30)+	0.425 (17)+	9.525 (381)+	11.40
10 µM	7.875 (315)+	0.575 (23)+	0.375 (15)+	8.825 (353)+	17.91
20 µM	6.425 (257)+	1.050 (42)+	0.250 (10)+	7.725 (309)+	28.14

^a^ Statistical diagnosis using the estimation of spot frequencies and confidence limits according to Frei and Würgler [41] for comparison with deionized water; +, positive; −, negative.

## Data Availability

The datasets used and/or analyzed during the current study are available from the corresponding author upon reasonable request.

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
