# Peer review of "The Toxicological Assessment of Anoectochilus burmannicus Ethanolic-Extract-Synthesized Selenium Nanoparticles Using Cell Culture, Bacteria, and Drosophila melanogaster as Suitable Models"

_nanomaterials, 2023, doi:10.3390/nano13202804_

Round 1

Reviewer 1 Report

Thisis a very complete study evaluating in vitro cytotoxicity and antigenotoxicity and antioxidant properties of a selenium-based nanoparticles (Anoectochilus burmannicus ethanolic extract-synthesized selenium nanoparticle (ABE-SeNP)), as an attempt do safety evaluation considering EFSA recommendation for testing safety of proposed novel foods or ingredients.

I consider that the manuscript is properly written, the methodology described clearly and precise, and Results described in results given an appropriate based of the conclusions.

I consider that these data are a good base for deciding further studies. However, authors present their conclusion as if these may be sufficient for supporting as novel food under European guide and regulation.

The sentence: “Taken together, it can be concluded that ABE-SeNPs may be safe to use in humans and animals and have non-cytotoxic effects when used at an effective dose” need to be modulated.

Authors had to be more prudent in their final conclusions.

Reviewer 2 Report

The work by Pensiri Buacheen, Jirarat Karinchai, Woorawee Inthachat, Chutikarn Butkinaree, Chonchawan Jankam, Ariyaphong Wongnoppavich, Arisa Imsumran, Teera Chewonarin, Nuttaporn Pimpha, Piya Temviriyanukul and Pornsiri Pitchakarn entitled “Toxicological assessment of Anoectochilus burmannicus ethanolic-extract synthesized selenium nanoparticle using cell culture, bacteria and Drosophila melanogaster as suitable models” is undoubtedly relevant, because selenium deficiency can cause disorders in almost all body systems (cardiovascular, endocrine, musculoskeletal, nervous, reproductive, etc.). According to recent data, over 1 billion people may have some form of selenium deficiency. The search for non-toxic food additives containing biologically available selenium is an important task in this field. In my opinion, this work was carried out at a high methodological level, may be of interest to readers of the journal Nanomatterials and can be published after some additions and corrections.

I believe that the introduction could be supplemented by adding references to the most recent review articles on the synthesis and application of SeNPs, as well as pointing out the additional benefit of SeNPs antimicrobial activity, which undoubtedly increases the attractiveness of the application of SeNPs and the food industry. For example, works doi: 10.3389/fmicb.2023.1229838, doi:10.3390/ma16155363, doi:10.3390/pharmaceutics15092253, ect. In the introduction, it is worth pointing out why kinsenoside is considered as the leading biologically active component of the Anoectochilus burmannicus extract.

Table 1. Was the distribution of NPs by size and zeta-potential unimodal? It is advisable to add a figure with distribution diagrams of SeNPs and ABE-SeNPs by size and zeta-potential.

Section 2.6. The passage numbers should be indicated for cell lines used in an experiment. Were micrographs of cells taken under different conditions (at least in transmitted light without staining)? If so, were any changes in cell morphology detected? I believe that adding such information will increase the attractiveness of the manuscript.

It is advisable to provide a brief description of the principle of hemolysis assay method.

If possible, it is also advisable to add photographs of examples of normal and mutant forms in the Drosophila mutagenicity test: small single spots, large single spots, and twin spots.

The PDI abbreviation should be decrypted on first use.

Symbols of statistical significance should be added. You can use different icons *, § etc. or letter designations a,b ect. The authors should explain why the figures show the 50% limit? Why was she chosen? The data on sample size and the test used to assess statistical significance should be added in the figure caption.

Figure 2, Table 3, Table 5. The information about the criterion used for assessing statistical significance should be added to the annotation.

Table 6. What were SD of frequencies of mutant spots? If possible, the information about statistical significance between ABE-SeNPs variants with positive controls should be added.

Cases of characters (such as H2O instead of H2O) need to be checked throughout the manuscript.

Best regards

Round 2

Reviewer 2 Report

I would like to recommend the authors some expand their methodological base in future researches. Assessing viability using fluorescence microscopy and/or flow cytometry can provide complementary information to MTT staining. For example, the proportion of cells in early phase of apoptosis or cells proportion in different phases of cell cycle can be estimated with fluorescently labeled Annexin V protein, fluorescent probes Hoechst and PI.